

# Detection of coffee leaf disease and identification using deep learning

Nameer Baht[1,*], Enrique Dominguez[1,*] and Saif Aljumaili[2]

[1] Department of Computer Science, Malaga University, Malaga, Spain
[2] Electrical and Computer Engineering, School of Engineering and Natural Sciences, Altinbas University, Istanbul, Turkey
* These authors contributed equally to this work.

## ABSTRACT

The sustainability of coffee production is a concern for many coffee-producing countries. Indeed, the conservation of the production of coffee needs to detect disease and infection in the early stages, to provide the ability to control and remedy. Coffee is one of the most consumed daily beverages, so it is considered one of the most important plant crops that affect the economy of the country that produces it. Thus, implementing systems for disease detection that do not require expert consultation can streamline production processes. In this paper, we proposed an efficient and rapid system utilizing cost-effective devices for detecting coffee leaf diseases to support farmers without the need for specialized expertise, leveraging deep learning models. Our technique involves several types of artificial intelligence (AI) models used, which include proposed new models using convolutional neural networks (CNN1, CNN2), and prompt transfer learning (VGG16, ResNet50, and EfficientNet), as well as applied machine learning supervised classifier with hybrid approaches (support vector machine (SVM) and Random Forest (RF)). In terms of training, the proposed model was fed with large datasets that contain five classes, with a total number of images exceeding 50,000 images, while the testing utilized a separate dataset. Finally, the results showed high performance across all evaluation metrics. CNN1 obviously distinguished that it has the superior accuracy compared to the other models, with a 99% value. In conclusion, the proposed framework has the capability to be applied, and it will deliver on-hand support to farmers.

## INTRODUCTION

Coffee is a plant that is consumed in large quantities every day. It also has an economic impact on daily coffee sales and international coffee prices. The spread of coffee-related diseases has a major impact on the growth of coffee production. The most dangerous diseases that affect coffee, such as Cercospora, Phoma, Rust, and Miner, have various causes, including weather conditions and unsuitable climates. Early identification of coffee diseases is crucial to prevent losses and enhance production and quality (*Karia, Ally & Leonard, 2025*). For instance, brown eye spot causes spherical necrotic lesions with a dark center and yellowish hue, leading to rapid leaf drop and dry branches, reducing plant output and fruit quality. Leaf Rust disease manifests as shapeless lesions, causing early

Corresponding author
Saif Aljumaili,
saifabdalrhman@gmail.com

leaves to fall and significant productivity reduction ranging from 30% to 55% in conducive weather conditions (*Aloyce, 2025*). To address these issues, developing an automated technique for detecting these infections in coffee leaves is recommended. The lack of disease structure necessitates a texture extraction process before disease recognition, utilizing features like factual highlights and nearby double examples. Recent advancements in convolutional neural networks have been explored for identifying coffee plant diseases. This study's key contribution lies in using trait vectors with limited data units for neural networks to classify images of coffee leaves instead of directly using image pixels.

Arabica and Robusta are the two main types of coffee globally (*Freitas et al., 2024*). Robusta coffee (Robusta) blooms in areas of the world with wetlands and at low altitudes in the eastern hemisphere in Africa, Indonesia, and Vietnam, which is the largest producer of this variety. Arabica coffee (Arabica) is cultivated in the uplands of Africa (*Kafle & Karki, 2025*). It is considered more resistant to diseases and constitutes about 30% of the coffee production in the world. Arabica is one of the finest coffees and is referred to as gourmet coffee. It is of higher quality than Robusta. Most varieties of Arabica beans are named after the country or region in which they were found or originated, and they make up the remaining 70% of coffee production worldwide. The level of caffeine in the Arabica coffee variety is about 1.5%. On the other hand, the level of caffeine in Robusta is 2.7%, which is almost double the level in Arabica. In general, Arabica has a higher price than Robusta, which is 1/3 lower than Arabica (*Alamri, Rozan & Bayomy, 2022*).

Nowadays, artificial intelligence (AI) plays an essential role in the development of object detection systems, and the field of disease detection is very important for countries whose economies depend on agriculture and coffee production. Deep learning methods surpass traditional machine learning approaches across various domains due to their exceptional performance. Among these methods, convolutional neural networks (CNNs) excel in image classification, object detection, and task identification. CNNs automatically learn relevant features from training data, unlike traditional methods that rely on manually crafted features based on prior problem knowledge. CNNs simplify the segmentation step through intrinsic convolutional filters, enhancing usability. They can play a crucial role in detecting plant diseases and issues. Despite challenges in automating plant disease diagnosis, the main contribution of this article is the development of deep learning models for an intelligent system capable of accurately classifying coffee leaf diseases using custom CNN models, as well as transfer learning-based models such as Visual Geometry Group 16 (VGG16), Residual Network 50 (ResNet50), and EfficientNetB0. The performance of hybrid models was also explored by combining CNNs with classical classification algorithms such as support vector machine (SVM) and Random Forest (RF). Implementation on low-cost devices and accessibility: Low-cost devices are more accessible and affordable, making advanced AI applications available to a broader audience, including in regions with limited technological infrastructure. In addition, the system for farmers and professionals to identify diseases easily, quickly, and accurately without expert consultation. The main objectives of the proposed system are to help countries whose economies depend on coffee production and limit the spread of diseases,

in addition to helping farmers in the early diagnosis and treatment of diseases and pests. The proposed system involves capturing images, which are processed by deep learning to display the plant's disease impact efficiently.

## Motivation

(1) To provide a new method that can handle a huge amount of datasets, as well as low image dimensions, whereas normally a high image to provide a perfect outcome.

(2) To propose a highly effective method for classifying coffee leaf disease that has the ability to classify with less complexity than the traditional methods.

(3) To support farmers, provide an easy-to-use and low-cost system that can be applied to mobile devices to enhance productivity and reduce losses resulting from diseases through early diagnosis.

(4) The proposed model is based on the urgent need for modern technologies in smart agriculture to improve coffee productivity and address disease challenges in a technologically advanced manner, contributing to improving the global agricultural economy. It is also a comprehensive and integrated solution to detect the most dangerous diseases affecting coffee plants accurately and quickly. Without the need for experts in this field to enhance productivity and reduce losses resulting from diseases through early diagnosis

## LITERATURE REVIEW

The challenges in detecting coffee diseases are still great and involve many factors, plant environmental complexities, and diversity in plant diseases, in addition to early diagnosis, which may be difficult to identify diseases and pests in their early stages. Therefore, these reasons greatly affect increasing production of the coffee crop. CNNs are motivated by the way that mammals use the layered architecture of their brains to represent their surroundings. A pattern recognition paradigm for computer vision developed in part because of these researchers' inspiration. *Novtahaning, Shah & Kang (2022)* worked on classifying and diagnosing four dangerous diseases that affect coffee leaves to limit their spread using pre-trained educational transfer. The proposed method was robust to noise, but the dataset used was not large. *Esgario, Krohling & Ventura (2020)* focused on images of Arabica coffee leaves using 1,747 images to classify the degree of risk and biotic stress of coffee diseases. The proposed multi-task system is based on VGG16 and ResNet50. The VGG16 model achieved better results and accuracy in identifying various viral diseases, with an accuracy of up to 95.47%. *Rangarajan Aravind & Raja (2020)* worked on the classification of ten different species by transferring images obtained from a smartphone to a computer *via* a local area network (LAN). Transfer learning is also used in four major agricultural crops that are the least explored. The images were classified in real-time and a GoogleNet based model, achieving the best result accurately of 97.3%, evaluated the prediction scores for each disease. *Marcos, Rodovalho & Backes (2019a)* proposed a neural network to detect coffee leaf rust diseases. The authors compared the results with different techniques, and they showed that a simple corrosion confirmation improved detection. The proposed method was able to identify the infection with high accuracy, but using a

small dataset consisting of 159 images provided by an expert. *Marcos, Rodovalho & Backes (2019b)* focused on coffee diseases in Brazil. Genetic algorithms were to calculate the optimal convolution kernel channel that emphasizes the texture and color characteristics of the infection, as the data provided by the experts indicated that this approach is a possible solution to the problem of rust identification (*Binney & Ren, 2022*). In this research of the selected dataset JMuBEN2, the data was trained by transfer learning three algorithms: ResNet50, Densenet-121, and VGG19. These models classified into three diseases: Phoma, Cercospora, and Rust. The Densenet-121 model outperformed other models and achieved an accuracy of 99.36% (*Taye & Goel, 2024*). The study addressed four types of diseases through the feed-forward model, and ResNet50 and the Inception V3, achieving an accuracy of 99.8% (*Hitimana et al., 2024*). In this article, a pioneering mobile application is presented, equipped with reporting capabilities supported by GPS to detect coffee disease detection system. Five models were tested (Xception, ResNet50, Inception-V3, VGG16, DenseNet), demonstrating DenesNet as having the highest accuracy 99%, 57% (*Yamashita & Leite, 2023*). This article clarified the processing of coffee leaves in Brazil by integrating two datasets containing 6,000 images in two stages using a MobileNet architecture achieving 98% (*Elezmazy, Abouhawwash & Mostafa, 2024*). This work demonstrated the use of a dataset from Kaggle (JMuBEN Coffee Dataset, https://www.kaggle.com/datasets/noamaanabdulazeem/jmuben-coffee-dataset) for two types of diseases using transfer learning techniques, and the model MobilelNet achieved accuracy 97%. Table S1 represents a comparison with modern methods used to detect coffee leaves with various diseases and pests and types of data sets.

We summarize the most important contributions in the proposed method; we propose to develop deep learning models for reliable disease detection with the diversity of diseases affecting coffee and increasing data. Furthermore, our approach utilized the CNN models that are fine-tuned using an initial hyperparameter setup, which can replace traditional disease detection in coffee plants and improve the overall classification accuracy processing involves the immediate analysis and generation of results as data is being received. In addition, the models are not only accurate and efficient but also capable of adapting to new data and improving over time. The integration of advanced feature extraction techniques and flexible voting further enhances the robustness of the models, making them suitable for a wide range of image classification tasks. Based on the experimental results provided on a large dataset, the models showed better results than other state-of-the-art approaches. Thus, the suggested models have the ability to implement low-cost devices that provide benefits for farmers and can easily detect diseases without the need to consult experts.

## MATERIALS AND METHODS

In this paper, we propose implementing the model using a comprehensive set of 11 algorithms belonging to different categories of deep learning techniques, including traditional deep learning algorithms, hybrid deep learning models, and transfer learning algorithms. These algorithms were carefully selected to evaluate the performance of the proposed model from multiple perspectives and achieve a rigorous systematic comparison between different strategies for classifying coffee leaf diseases. The proposed methodology

consists of three stages. The first stage involves selecting large and important data sets containing the most common diseases affecting coffee crops, which were collected by specialists in this field. The images were resized to dimensions (100 × 100 × 3), and the data were distributed equally using the random undersampling technique. The minimum number of samples was determined in the smallest class, which results in an equal distribution across all classes by removing samples from the larger classes. Then, Min-Max Normalization was used to distribute the numerical values within a specific range (0, 1) to improve the performance of the algorithms, speed up the training process, and reduce problems resulting from differences in measurement units. Augmentation is the step that follows normalization, in which we used five methods: flipped left-right (lr), flipped up-down (ud), rotated, brightened, and blurred. These processes aim to simulate the diversity of real data and improve the model's ability to generalize. Moreover, two techniques were used. Local binary patterns (LBP) was used to extract texture features from the images. LBP is particularly effective in capturing the local spatial patterns, which are crucial for distinguishing fine details within the images. Extended center-symmetric local binary pattern (XC LBP) further enhances the feature set when applied to extract more robust and discriminative features. This technique builds upon LBP by fogging it more resilient to noise and variations in lighting. The second stage involves splitting the data for training and testing after processing to implement the proposed algorithms, which include a comprehensive set of different deep learning algorithms (CNN1, CNN2, and Transfer learning, VGG16, ResNet50, efficient, and hybrid learning, support vector machine, Random Forest).

The third stage: Flexible Voting: To improve overall accuracy, a flexible voting mechanism was adopted. This method combines the output of all models, allowing for more reliable and robust predictions. The outputs are then used to diagnose and classify the type of disease with high accuracy.

Deployment and interface development model saving after successful training, the models saved for deployment user Interface, a user-friendly interface was developed, enabling users to upload images easily and receive instant predictions. The interface is optimized for quick processing, ensuring minimal latency between image upload and prediction results a dedicated program was developed to interface with an external camera, allowing real-time image capture captured images are quickly processed and saved, providing a continuous stream of data that can be used for future retraining and model improvement can be used for future retraining and model improvement. Figure 1 shows the flowchart of the work.

## Data preparation and preprocessing

JMuBEN and JMuBEN2 (Arabica dataset): The dataset was collected by disease detection experts and supervised by the dataset is supervised by three institutions: Chuka University, the University of Embu, and Jomo Kenyatta University of Agriculture and Technology. The first group contains three disease categories: Cerscospora, Leaf Rust, and Phoma. The second group contains two types of diseases: Miner and Healthy. The two volumes contain (58,555) images, with a size was 128 × 128 pixels for five classes (Phoma, Cercospora, Rust,
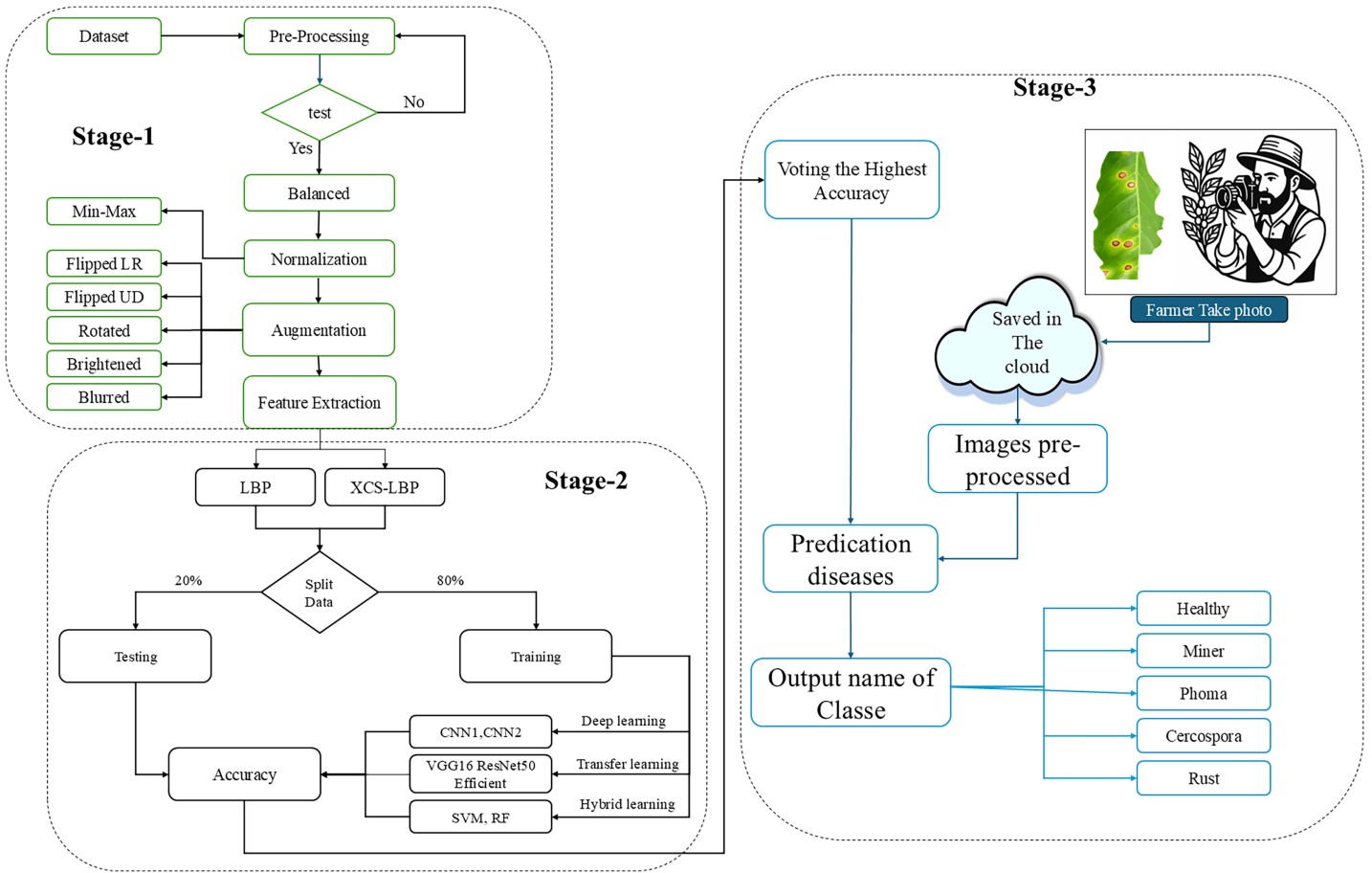

**Figure 1 Overview of the method.** Stage 1 is the data acquisition and applies several types of preprocessing to prepare the dataset; Stage 2 is the feature extraction form the images and implements the models (training and testing) several types of methods, Stage 3 is the identification of diseases by applied from the farmer.

Healthy, Miner). The dataset used for training was obtained from publicly available online sources. The source is open access and widely used in research studies related to the detection of coffee disease (*Jepkoech et al., 2021*).

Dataset collection is an important part of the building blocks of a system that detects plant diseases, especially coffee. Experts in the field determine the type of disease, collect data, and the challenges associated with collecting data are many, such as weather, accuracy, and other important things (*Yebasse et al., 2021*). Coffee disease experts classified each image accurately; the analysis focuses on the identification and quantification of various coffee leaf diseases. The dataset comprises images of coffee leaves, categorized into different disease types. The primary objective is to classify and count the number of leaves affected by each disease type. The data underscores the importance of monitoring and addressing coffee leaf diseases to ensure crop health and productivity. The significant number of healthy leaves is promising, yet the presence of diseases like Miner and Leaf Rust highlights areas requiring attention and potential intervention. The test dataset is then used to show how well the recommended architecture can summarize the new

**Table 1 Types of the most common coffee diseases.**

| Type of leaf | Number of images | Description | Images |
|---|---|---|---|
| Healthy | 18,985 | Green without any spots or damage of any kind. Large, wavy dark patches on the leaf's upper surface | Table_2_Picture1_Healthy.jpg |
| Miner | 16,979 | Rubbing an area or bending a leaf causes the upper epidermis to break, revealing tiny white caterpillars in the new mines | Table_2_Picture2_ Miner.jpg |
| Phoma | 6,572 | A leaf that turns brown and dies starting from the tip area. | Table_2_Picture3_ Phoma.jpg |
| Cercospora | 7,682 | Dry areas that are brown in color with a border in the shape of a bright halo around it | Table_2_Picture4_ Cercospora.jpg |
| Rust | 8,337 | Features patches that resemble a halo that ranges in color from yellow to brown. | Table_2_Picture5_ Rust.jpg |

information or to show how well the model performs under real conditions. Datasets were obtained from a plantation of Arabica coffee with the help of a camera and a plant pathologist. The images were then edited through cropping focusing on parts of interest, along with samples and numbers for each of them. Image augmentation was done to increase the dataset size and prevent overfitting challenges during the training and validation of the model. Table 1 shows the types of diseases and a description of each disease, along with samples and numbers for each disease.

In addition, we selected another dataset to test the proposed model and generalize it. The dataset is open source data from the Kaggle platform called coffee leaf disease. It containing 1,700 images, divided for training and testing. The dataset was specifically designed for detecting coffee diseases, consisting of three channels with a resolution of 2,048 × 1,024 pixels. It includes four types of diseases (Miner, No Disease, Phoma, and Rust). This dataset focuses on the problem of detecting healthy and diseased coffee leaves (*Murhaban & Suhendra, 2023*).

Figure 2 shows the image illustrates three basic steps in data processing used for coffee leaf classification. Original images: The image selected from the dataset contains significant variations in lighting, color, and the number of images per disease class, demonstrating an imbalance in the number of samples across classes. After balancing: Data balancing techniques, such as oversampling for underrepresented classes, were applied to achieve an even distribution of images across all classes. This step is important in reducing bias in the proposed model and improving its accuracy. After normalization: The images then undergo a normalization process, where pixel values are converted to a uniform numerical range, such as 0 to 1. This helps the model learn better from the images and reduces the impact of lighting variations or color contrast.

To address the class imbalance, we balanced the dataset, resulting in an equal number of samples for each class. We used algorithm random undersampling balance data undersampling is a technique used to balance an imbalanced dataset by reducing the number of samples in the majority class. This method helps ensure that the model does not become biased toward the majority class during training. Each class now has around 6,572 samples. This balancing ensures that the machine-learning model receives an equal representation of each class, thereby improving its ability to generalize across different

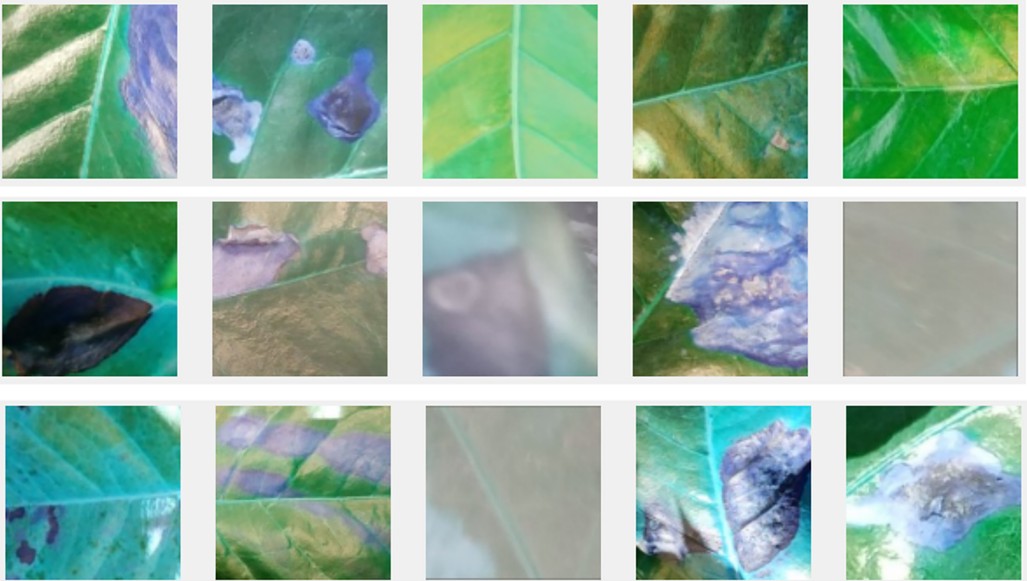

**Figure 2  Sample images of the method that were applied after processing the whole dataset.**

disease types. The main steps for undersampling are to: identify the majority and minority classes, randomly remove samples from the majority class, and combine the dataset. These images demonstrate a variety of diseases, highlighting the diversity and challenges in classification balancing the dataset to address the class imbalance each class was limited to a minimum count of 6,572 images. This approach ensured equal representation of each class, improving the model's fairness and performance. The images were normalized to have pixel values between 0 and 1 by dividing by 255.0. This normalization step is crucial for ensuring consistent input ranges for the machine-learning model. Labels were converted to one-hot encoding using label binaries, transforming categorical labels into a binary matrix format; this is essential for multi-class classification tasks. The preprocessed dataset was split into training and testing sets using an 80–20 split: training data shape: (26,284 samples); testing data shape: (6,571 samples). Figure 3 represents the number of samples in each class before and after balancing. The classes are evenly distributed, which should help improve the performance and accuracy of our mode.

To enhance the model's generalization ability and increase its robustness, a set of extension techniques was used to improve image diversity and reduce the likelihood of overgeneralization. These techniques included flipping the images horizontally and vertically, rotating them using affine transformations at an angle of 20 degrees, adjusting the brightness by 1.2, and applying a Gaussian blur with a standard deviation ($\sigma$) of 1.0, Table 2 shows all the lists the mathematical operations for the augmentation techniques. These operations aim to simulate the variability of real data and improve the model's generalization ability. These manipulations aim to simulate realistic changes in leaf
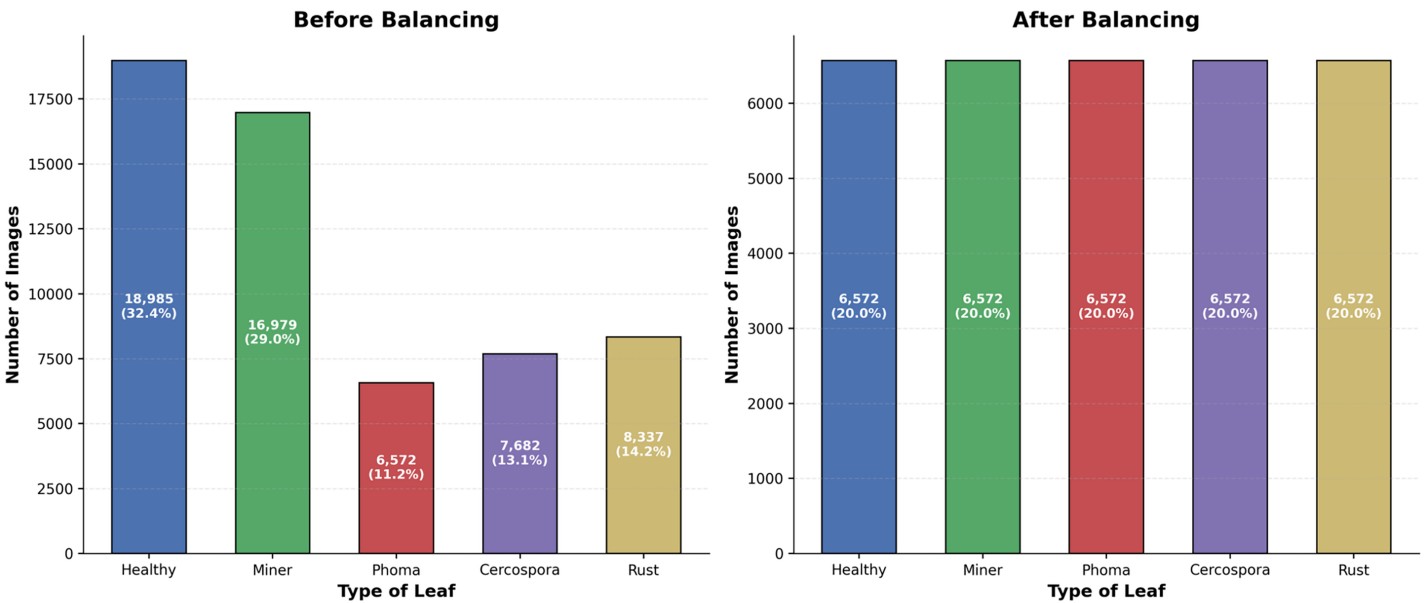

**Figure 3** Sample images before and after balancing.

**Table 2 The mathematical operations for the augmentation techniques.**

| Technique | Mathematical equation |
|---|---|
| Horizontal flip | $I'(x + y) = I(w - 1 - x, y)$ |
| Vertical flip | $I'(x, y) = I(x, H - y - 1)$ |
| Rotation | $\begin{bmatrix} x' \\ y' \end{bmatrix} \begin{bmatrix} \cos(\theta) & -\sin(\theta) \\ \sin(\theta) & \cos(\theta) \end{bmatrix} \cdot \begin{bmatrix} x - x_c \\ y - y_c \end{bmatrix} + \begin{bmatrix} x_c \\ y_c \end{bmatrix}$ |
| Brightness increase | $I'(x, y) = \alpha.I(x, y) + \beta$ |
| Gaussian blur | $I'(x + y) = \sum_{i=-k}^{k} \sum_{j=-k}^{k} I(x + i, y + j).(i, j)$ |

appearance, improving the model's ability to detect diseases under different environmental conditions (*Maharana, Mondal & Nemade, 2022*). Figure 4 shows augmentation for five categories of diseases with the approaches utilized.

## Feature extraction

**Local binary patterns (LBP)**: Feature extraction is a crucial step in many machines learning and image processing tasks, including your coffee disease detection project. It involves identifying and extracting important and relevant features from raw data to improve the performance of models from color features, texture features, LBP and others. CNNs automatically learn complex features directly from the image data, often outperforming traditional handcrafted features. In addition to preprocessing, feature extraction was performed to enhance the analysis. Grayscale image and LBP. The grayscale version of the image is created to reduce computational complexity and highlight texture details. LBP applied grayscale images to capture texture features. LBP is a powerful feature

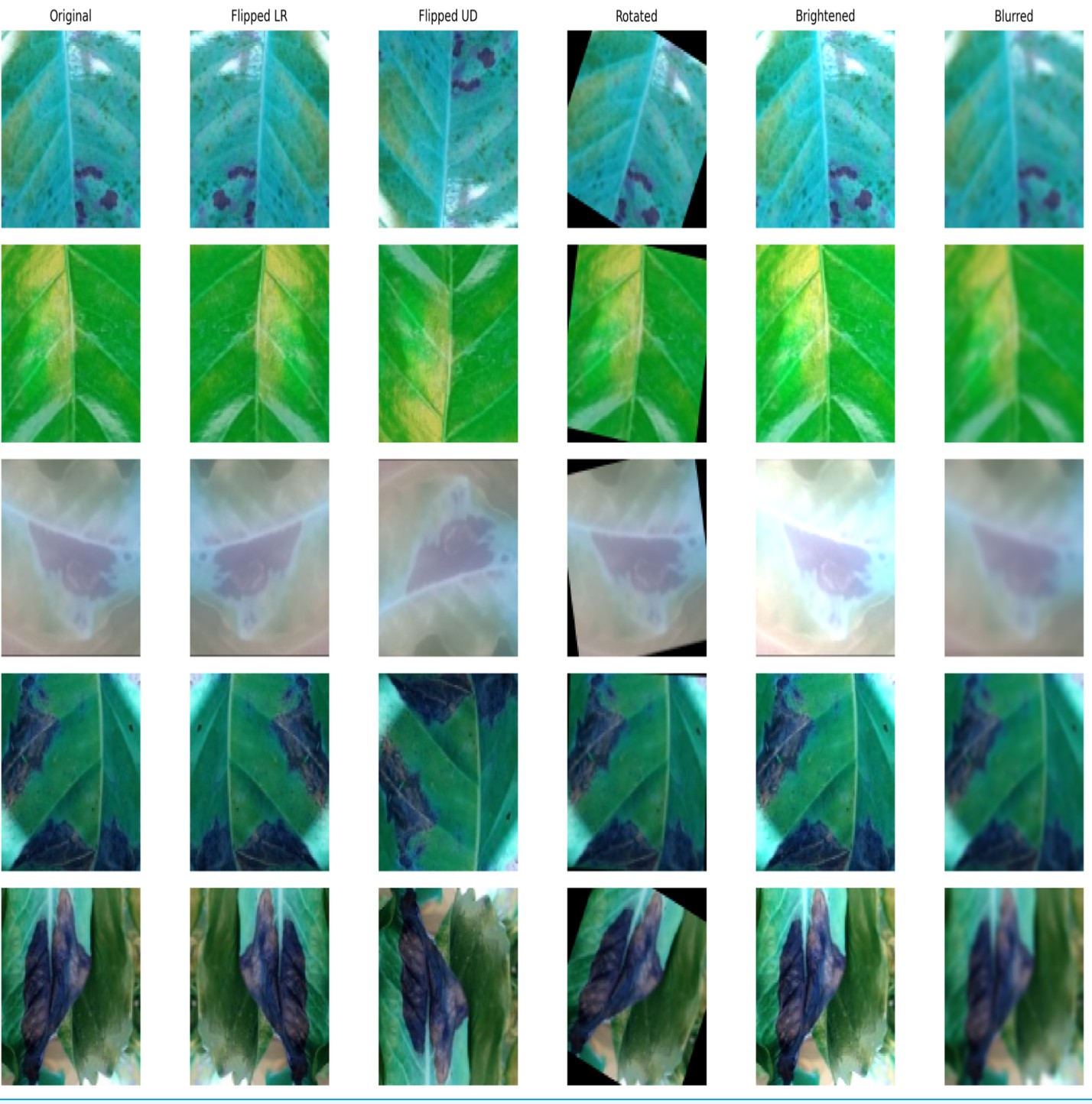

**Figure 4 Images from the augmentation dataset.**

descriptor that provides the spatial structure of local image texture. The histogram of LBP values computed to quantify the texture information. This histogram is used as a feature vector for classification. Figure 5 shows an example of the application of the LBP technique.

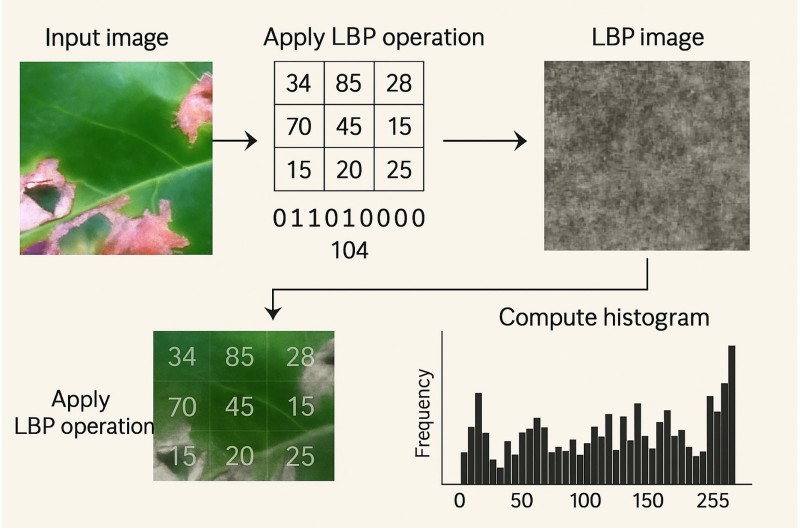

**Figure 5** **Example of application of the LBP technique to a sample image.** The histogram is the feature vector used for classification.

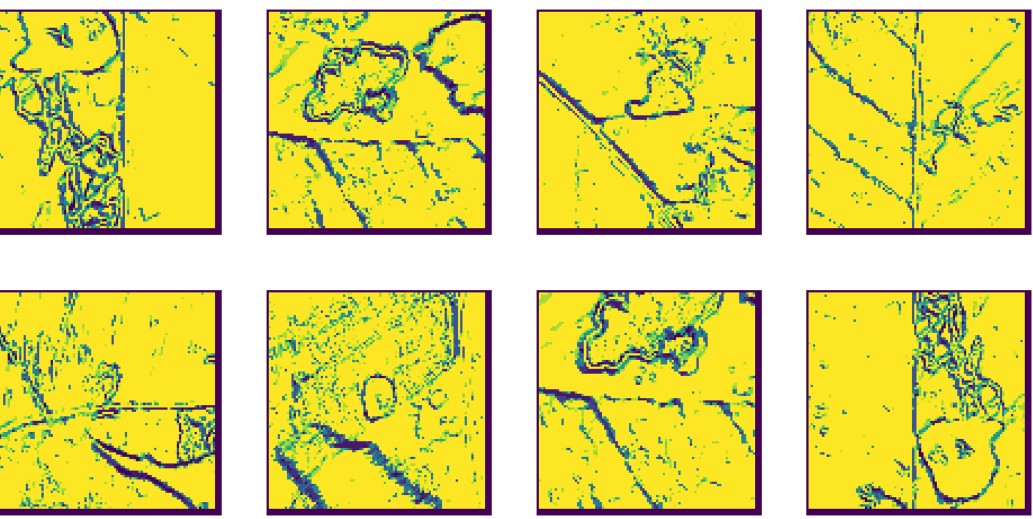

**Figure 6** **Extended center-symmetric local binary pattern (XCS-LBP) feature extraction.**

**Extended center-symmetric local binary pattern (XCS-LBP)**: XCS-LBP applied further to capture more detailed texture information. This technique helps in highlighting finer features of the leaf surface, which are crucial for disease identification LBP used to ex-tract texture features from an image, each pixel's neighborhood encoded into an LBP histogram, which represents the texture information classification. The XCS algorithm applied to these LBP histograms to classify the images or patterns. The XCS system evolves classifiers based on the features extracted by LBP, improving its classification accuracy over iterations, XCS-LBP leverages LBP for extracting texture features and combines it with the

XCS algorithm for effective classification, making it a powerful tool for pattern recognition and image analysis and all these processes are shown in Fig. 6.

## CNN models

CNN architectures specifically designed for image analysis and object detection (*Khushi et al., 2024*). These architectures often consist of multiple convolutional layers followed by pooling layers to extract features from images. The final layers usually consist of fully connected layers for classification (*Biswas, Saha & Deb, 2024*). Activation functions are a critical component of CNNs and used in the introduction of nonlinearity into each layer is output. Most real-world data is complex and cannot be modeled well with a linear function. Non-linearity is therefore important (*Perumal et al., 2024*). There are several activation functions used in CNNs, but some of the most popular ones are rectified linear unit (ReLU) (*Rastogi, Dua & Dagar, 2024*). This function is simple and computationally efficient in setting all values that are negative to zero leaving all values that are positive unchanged.

The CNN1 architecture was built using a sequential stack of layers. The CNN designed in this model consists of a set of concatenated layers that aim to extract features from input images ($100 \times 100 \times 3$), and classify them into five classes. We divided the data into 20% test and 80% training. The input layer receives color images of size $100 \times 100 \times 3$. The first convolutional layer (Conv2D) contains 32 filters of size $3 \times 3$ with a ReLU activation function. The first pooling layer (MaxPooling2D) reduces the dimensionality to $49 \times 49 \times 32$ using $2 \times 2$ pooling. The second convolutional layer (Conv2D) applies another 32 filters of size $3 \times 3$ with ReLU. This is followed by the second pooling layer (MaxPooling2D), which reduces the size to $23 \times 23 \times 32$. The dropout layer then drops 50% of the units to reduce the possibility of overfitting. The flatten layer transforms the output from 3D to a vector of size $23 \times 23 \times 32$. The fully connected layer (Dense) contains 256 units with ReLU. The second dropout layer is responsible for aggregating the discovered features, and it drops 70% of the units during training to reduce overfitting. It has no parameters. The output layer (Output Dense) has five neurons (one for each class) and uses SoftMax to give the probability of each class, epochs = 5, with Adam being one of the most widely used optimizers. It combines the advantages of SGD and RMSProp. It automatically adapts the learning rates for each weight.

The second model CNN2 is built on a deep Conv2D neural network architecture, consisting of three consecutive Conv2D layers, each followed by a MaxPooling 2D pooling layer, with two batch normalization layers added to improve training stability. The model starts with an input layer of size $100 \times 100 \times 3$, followed by a first convolutional layer containing 25 $5 \times 5$ filters with "same" padding. This is followed by a second convolutional layer after pooling, containing 50 $5 \times 5$ filters. The first batch normalization layer is applied to 50 channels. The third convolutional layer contains 70 $3 \times 3$ filters. Then, after a series of convolutional layers, a second batch normalization layer is applied. The output is transformed into a vector using a flatten layer. The data is then passed through two fully connected (Dense) layers. A dropout of 0.5 is applied to reduce overlearning. Finally, the

output layer contains five classification units with a SoftMax function, a batch size of 32, epoch of 5, and Adam optimizer.

The VGG16 architecture is pre-trained on ImageNet but excludes its top fully connected layers. The VGG16 model is loaded with weights and configured to accept input images of size $100 \times 100$ with three color channels. After loading the pre-trained VGG16 model, custom layers were added on top. The output from the VGG16 base model is flattened to convert the 2D feature maps into a 1D vector. Following this, three dense layers were added, each with ReLU activation. The first dense layer has 46 neurons, the second has 40 neurons, and the third has 10 neurons. The final layer is a dense layer with a number of neurons equal to the number of classes in the target data, using soft-max activation to produce class probabilities. The complete model is then created by specifying the VGG16 input and the custom output layers. The model was compiled using the Adam optimizer, categorical cross-entropy as the loss function, and accuracy as the performance metric. The total number of layers in the model is 23 layers. The optimizer used in the model is Adam.

Hybrid Learning: In addition to designing three basic models using convolutional neural networks (CNN1, CNN2, and VGG16), a set of hybrid models was developed to combine the power of deep learning in visual feature extraction with the effectiveness of traditional machine learning algorithms in classification. The final classification layer (Softmax) was removed from CNN1, CNN2, and VGG16, and their final hidden layers were used to extract deep feature representations of images. These features were then passed as input to two traditional classifiers: support vector machine (SVM) and Random Forest (RF). This process resulted in six hybrid models: CNN1 + SVM, CNN1 + RF, CNN2 + SVM, CNN2 + RF, VGG16 + SVM, and VGG16 + RF. The main idea behind this approach is to evaluate the extent to which traditional classification algorithms benefit from the deep features extracted by convolutional networks and compare their performance with direct classification results *via* the Softmax layer.

In addition to the previous models, two transfer learning algorithms were included: ResNet50 and EfficientNetB0, which are deep models pre-trained on the ImageNet dataset. These models were reused without modification to their basic architecture, except for replacing the final layer with a new one corresponding to the number of classes in the coffee leaf data. The goal is to evaluate the actual performance of these two models when used as baseline transfer learning, and to compare their results with those of existing and hybrid models. Neither ResNet50 nor EfficientNetB0 used traditional SVM or RF classification algorithms, relying solely on direct prediction *via* the final Softmax layer.

In the process of disease detection in plants, the classification step using image processing and computer vision is very important (*Upadhyay et al., 2025*). The proposed system was designed based on the process of detecting and extracting features in the affected leaf. The model's parameters are modified so that it correctly identifies and classifies images, and then the model is evaluated. After the training is completed, the model's performance is evaluated using the test dataset (*Ding et al., 2024*). The accuracy of the model and its ability to recognize diseases in health are measured. After verifying the model's performance, it can be used to classify new images of coffee plants and identify

**Table 3 Evaluation matrix of the trained model to the untrained dataset.**

| Classes named | | CNN1 | | | | |
|---|---|---|---|---|---|---|
| | | Predicted class | | | | |
| Actual class | Class 1 | 1,325 | 130 | 0 | 0 | 0 |
| | Class 2 | 0 | 1,297 | 0 | 0 | 0 |
| | Class 3 | 0 | 0 | 1,301 | 0 | 0 |
| | Class 4 | 1 | 0 | 0 | 1,266 | 0 |
| | Class 5 | 0 | 0 | 0 | 0 | 1,330 |

various diseases based on the patterns it learns (*Shrma & Rahiman, 2024*). We implemented the proposed method using Python, where the application was *via* the cloud using Jupiter Notebook, where libraries such as Pandas, NumPy, Matplotlib, and Seaborn were called to work with data consisting of images. In addition, the data was requested from the Kaggle website.

# EXPERIMENTAL RESULTS

## Hardware specifications used during experiments

The proposed model was implemented on the Kaggle platform, which provides a suitable cloud training environment for efficient model development and testing. The proposed CNN1 model was designed to be lightweight in terms of computational complexity, making it suitable for implementation on low-cost devices. Hardware specifications used during experiments: (1) Platform: Kaggle (Notebook Environment). (2) graphics processor (GPU): NVIDIA Tesla P100 (16 GB VRAM). (3) Memory (RAM): 13 GB). (4) Processor: Intel Xeon (approximately 2.3 GHz). (5) Software Environment: Python 3.10—TensorFlow 2.x.6). In this section, we provide a detailed explanation of the metrics used to evaluate the performance of the models. Moreover, the performance of the proposed models is also given and compared to other models published in the literature. Our experimentations focused on the performance and the detection classification.

To conduct a comprehensive analysis of the performance of the proposed models, confusion matrices were created for all proposed algorithms and presented in a unified table. This representation allows for comparison of the behavior of different models in classifying each category, and for identifying categories that exhibited confusion or misclassification across multiple models. The main diagonal displays the number of correct predictions for each category, while the other values represent the number of errors between categories. This detailed analysis helps explain differences in model performance, especially when overall accuracy values are close. Table S2 shows the confusion matrix for all algorithms.

To analyze the generalization ability of the proposed model, a new dataset consisting of approximately 1,700 images was tested using the highest-accuracy model (CNN1). The confusion matrix in Table 3 shows the classification results on this dataset, with an overall

**Table 4  Performance results on an independent test set using the proposed model.**

| Class | Accuracy | Average accuracy | Precision | Average precision | Recall | Average recall | F1-score | Average F1-score |
|-------|----------|------------------|-----------|-------------------|--------|----------------|----------|------------------|
| Class 0 | 97.50 | 98.0 | 99.50 | 99.48 | 97.50 | 98.7 | 98.50 | 99.09 |
| Class 1 | 96.00 | | 98.00 | | 98.00 | | 98.00 | |
| Class 2 | 98.49 | | 100.00 | | 98.00 | | 98.99 | |
| Class 3 | 99.00 | | 99.90 | | 100.0 | | 99.95 | |
| Class 4 | 99.00 | | 100.00 | | 100.0 | | 100.0 | |

accuracy of 98%. Class balance was maintained by randomly adding a fifth class from the training data, which helped maintain the consistency of the distribution.

This section, we provide detailed experimental results presented for the proposed models, which include convolutional neural networks (CNN1, CNN2, and VGG16), as well as hybrid models (CNNs with SVM and Random Forest), and transfer learning models (ResNet50 and EfficientNetB0). The proposed approach demonstrated that all models performed well, with accuracy ranging between 93% and 99% on the coffee leaf test dataset. CNN1 achieved the highest accuracy among the models, demonstrating its ability to efficiently extract distinctive visual features. Table S3 shows the results of each model for each class, with the models evaluated using four basic metrics: Accuracy, Recall, Precision, F1-score, and the average for each metric.

To demonstrate the comprehensiveness of the proposed approach, the model was tested on a second external dataset of 1,700 images distributed across only four disease classes, to evaluate its ability to generalize in a new data environment. To achieve class balance in the test set, images for the fifth class (Cercospora) were randomly called from the training data, so that the number of samples in each class was approximately equal. The results showed that the model maintained high performance, with an overall accuracy of 98% on this dataset, demonstrating the model's effectiveness in handling previously unseen data and its ability to reliably distinguish between different types of diseases. Table 4 shows the results we obtained by testing the second data set. The results show very high accuracy, which proes that the proposed approach is reliable and effective.

## Performance assessment

Developing an adequate system for the detection of coffee leaf diseases requires the accurate classification of different plant abnormalities. To achieve this, we performed a series of experiments, which were conducted in two stages. In the first stage, we focused on the classification performance of the proposed CNN models. This performance is also observed, where the proposed CNN models perform very well across all standard metrics, the confusion matrix shows a very high accuracy across all classes, with only With a few errors' misclassifications out of thousands of samples. The models perform exceptionally well in classifying the different categories of coffee leaf diseases, as evidenced by the near-diagonal matrix, indicating accurate predictions aligning with the true labels.

The receiver operating characteristic (ROC) curve was used to evaluate the performance of the trained models in classifying the five categories of coffee leaf diseases. The ROC

curve resembles a graph showing how well a classification model performs. The curve is interpreted to understand how the model makes decisions at different levels of certainty. The curve shows the relationship between the true positive rate and the false positive rate for each category. Multiple ROC curves were calculated for each of the proposed models and for the hybrid models, allowing for comparison of their performance in distinguishing different categories. The values and results in the auv indicate that the approach is effective on the test data, demonstrating a high ability to classify healthy and diseased leaves. Figure 7 shows the curves for each algorithm.

Deep learning is a powerful machine learning technique that you can use to train powerful object detectors (*Maruthai et al., 2025*). There are several deep learning techniques for object detection, including VGG16 to identify the area affected by diseases and determine its type (*Karthik, Alfred & Kennedy, 2023*). The performance values for the proposed methods to detect coffee leaf diseases are presented in Table 4. In this case, CNN1 has shown the best option to be implemented in a low-cost device for the detection of coffee leaf diseases.

CNN models have proven effective in classifying coffee diseases with high accuracy and speed detection. The performance of the proposed detection systems, built on the CNN1 and VGG16 architectures, exhibited a notable convergence in their effectiveness. Both architectures consistently delivered high accuracy in detection tasks, demonstrating their robustness and reliability. This convergence highlights the complementary strengths of the two algorithms, as they each successfully captured and processed intricate features within the dataset, leading to precise and consistent detection outcomes. Finally, a comparative study with other state-of-the-art methods was made, whose results are As previously mentioned in the introduction, there are many other methods published in the literature, but most of these studies were made using a relatively small size of datasets, and therefore those results are not very convenient for a real-world application. This more exhaustive study, using a large variety of real images and diseases, gives more reliability to the results achieved by the proposed methods for a real application.

## Training and testing time models

Figure 8 shows a comparison of training and testing times for different models. The differences between models is shown in terms of time efficiency. This means that the left axis displays the training time, and the right axis displays the testing time, both using different metrics on the same graph. The left blue axis represents the training time in seconds (s). Each blue bar represents the time it took the model to learn from the data. The right green axis represents the testing time in seconds (s). The green barsrepresent the time it took the model to predict outcomes on new data. Figure 8 depicts the training and testing time performance of the proposed models.

The VGG16 model with Random Forest and CNN achieved the shortest training and testing time, reaching 44 and 2.2 s, making it the most computationally efficient model. In contrast, the EfficientNet model required the longest training time, at 199 s, although its testing time remained within the acceptable range (3.5 s). Hybrid models that combine neural networks (such as CNN and CNN2) with traditional machine learning algorithms

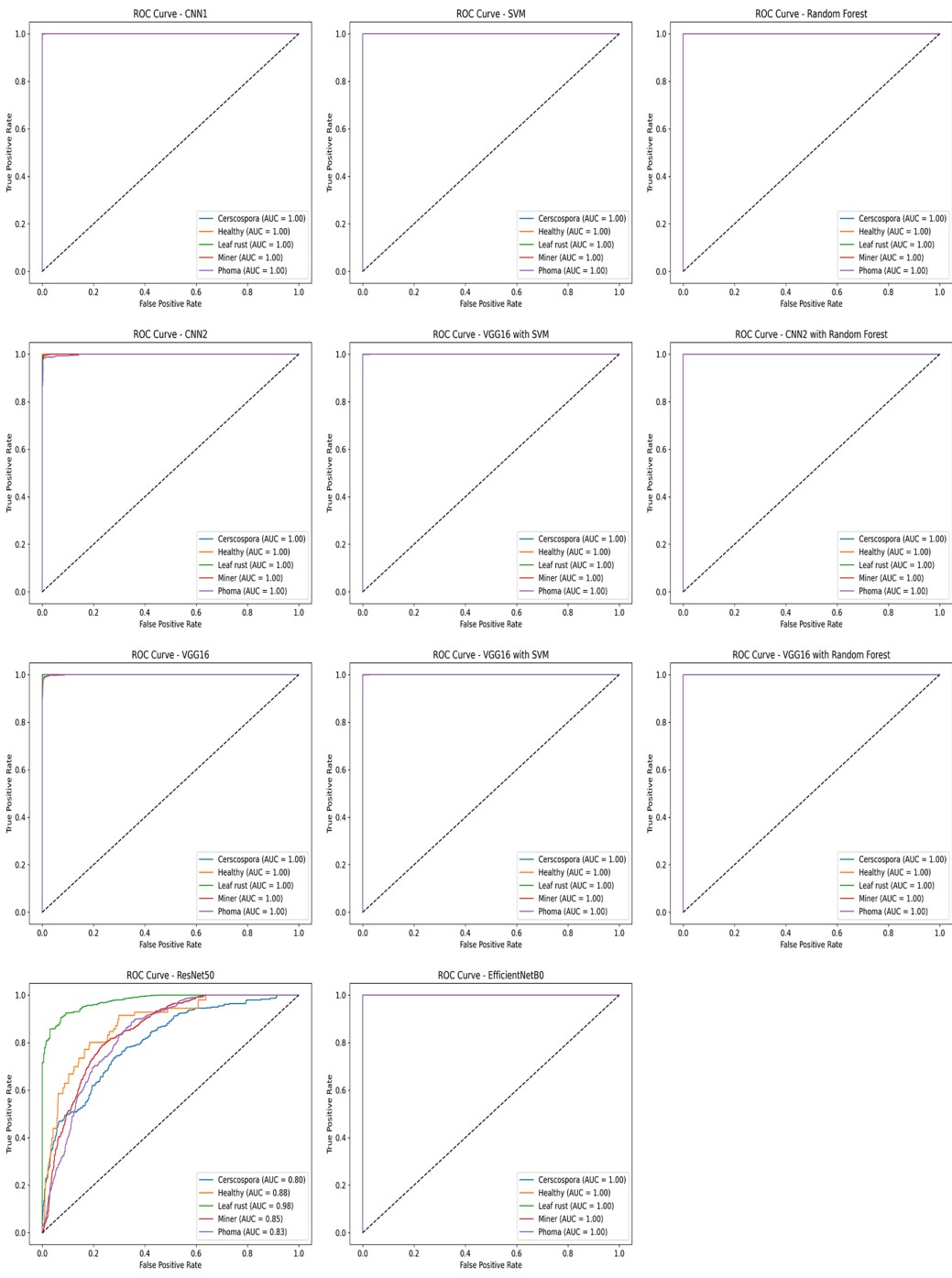

**Figure 7** ROC curves illustrating the performance of various classification approaches applied in this study.

(such as SVM and RF) demonstrated a good balance, such as the CNN2 + RF model, which delivered fast performance in training (122 s) and testing (2.5 s). These results demonstrate that hybrid models and convolutional neural networks can be an effective choice when real-time processing speed-dependent applications are required. Table 5 illustrates this.

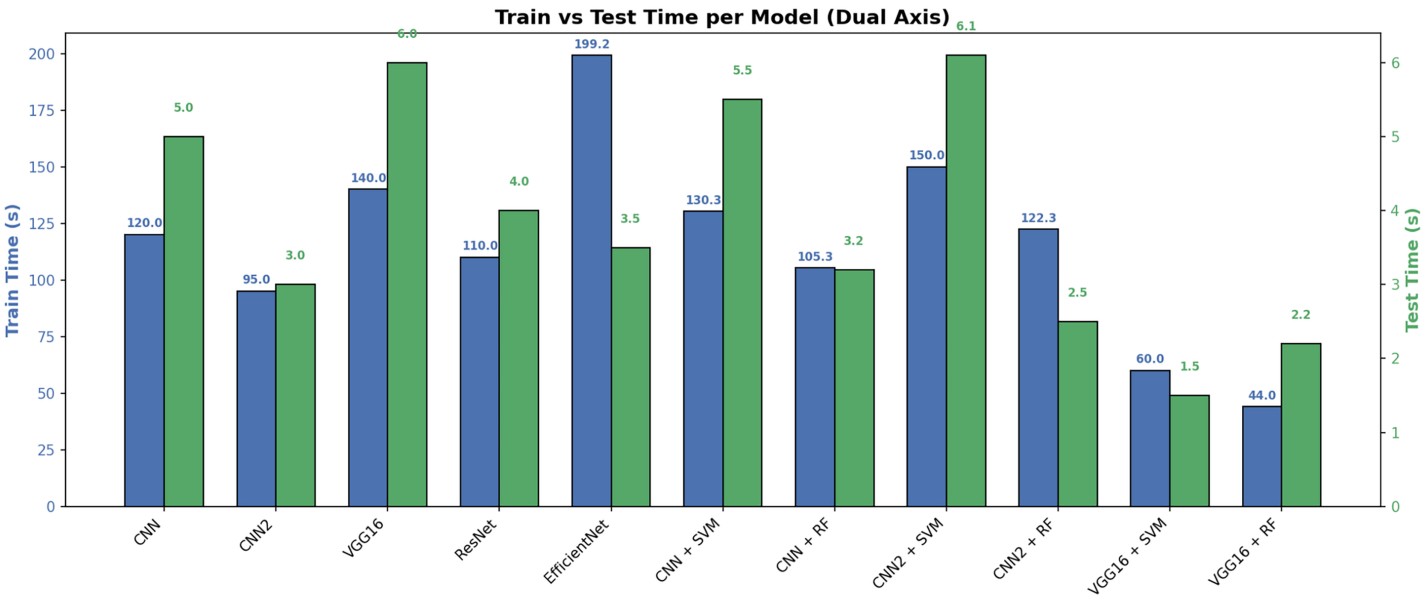

**Figure 8  Training and testing time per models.**

After training, the model uses a dataset containing images of coffee leaves classified into five classes (Intact, Rust, Phoma, Miner, and Cercospora), the trained model was used to predict which class of new images have never been seen belonging to before, as shown in Fig. 9.

A visual comparison of the model's interpretation decisions for five images of plant leaves is presented, using different interpretation techniques. Each row represents an interpretation of one image: the original image that was classified. The first technique is gradient-weighted class activation mapping (Grad-CAM), which focuses on the regions that the model focuses on in making its decision. The second technique is Grad-CAM++, which is an improved version of Grad-CAM, which focuses on reviewing features in the image for more than one object. Local interpretable model-agnostic explanations (LIME) is used to show the boundaries that most influence the model's prediction using local segmentation. The fourth technique is the SHapley Additive exPlanations (SHAP) bar plot, which illustrates each feature, such as color, texture, or a specific pixel location, for prediction. Analysis of the results on the selected images represents the first and second rows, representing clearly affected images. Grad-CAM++ and LIME highlight the affected area, while SHAP highlights important features that contribute to the prediction. This indicates that the model focused on diseased areas, not random ones, indicating good interpretation. Similarly, rows three and five show healthy images. Heat maps are low in concentration, almost empty. SHAP bar plots appear absent or have weak effects. This indicates that there are no strong signs of disease, which is positive. However, row four shows a leaf with a mild effect. Grad-CAM++ and LIME identify a specific area, while SHAP shows variations in the importance of features. Figure 10 shows the model's interability of the model to concentrate on the region of the diseases.

| Table 5 Model-wise computational in seconds for all models. | | |
|---|---|---|
| **Model** | **Train time (s)** | **Test time (s)** |
| CNN1 | 120 | 5.0 |
| CNN2 | 95 | 3.0 |
| VGG16 | 140 | 6.0 |
| ResNet50 | 110 | 4.0 |
| EfficientNet | 199 | 3.5 |
| CNN1 + SVM | 130 | 5.5 |
| CNN1 + RF | 105 | 3.2 |
| CNN2 + SVM | 150 | 6.1 |
| CNN2 + RF | 122 | 2.5 |
| VGG16 + SVM | 60 | 1.5 |
| VGG16 + RF | 44 | 2.2 |

## DISCUSSION

Despite the promising results of using CNNs for detecting coffee diseases, there are an outstanding number of challenges that require addressing (*Silva et al., 2025*). Some of the challenges are the lack of standardized datasets, the need for robust models that can generalize to different coffee varieties and environmental conditions, and the high cost of data labeling and model training. This prevents many of the approaches proposed in the literature from further generalization, as they rely on the bias extreme of the bias-variance tradeoff, cannot take advantage of a larger number of training images, and are unable to discriminate against diseases with subtle perceptual differences. One of the reasons for this is that classical algorithms have always been difficult to apply. Especially in the task of leaf disease classification, where several challenges must be addressed related to the high variability of symptoms for a specific disorder caused by different disease stages, multiple simultaneous disorders affecting the same leaf, the existence of similar visual symptoms in different diseases, or even the complexity on the acquisition of images, annotations, and quality. The most pragmatic solution to address these problems is to gradually enrich the diversity of the dataset. CNN models can be extended and adapted in complexity to match the expressiveness required by any given task, such as coffee plant disease identification, and data availability. In this work, a huge effort was made on the usage of large publicly available datasets such as JMuBEN and JMuBEN2 (Arabica dataset), which contain more than 58,000 images of coffee leaves with annotations regarding the state of the leaves and the disease. Moreover, to be sure about the results that we obtinaed, and for reliability and generalization the proposed method, we utilized another dataset for testing that conten (1,700) images, similar to the diseases on which we trained, our approach proved to be very accurate in detecting infections that appear on coffee leaves, which confirms the effectiveness of the proposed approach in dealing with real-world data that has not been previously seen.

Another challenge is the accessibility of the proposed tools in the literature. Many computerized methods have been developed for the automated detection and classification
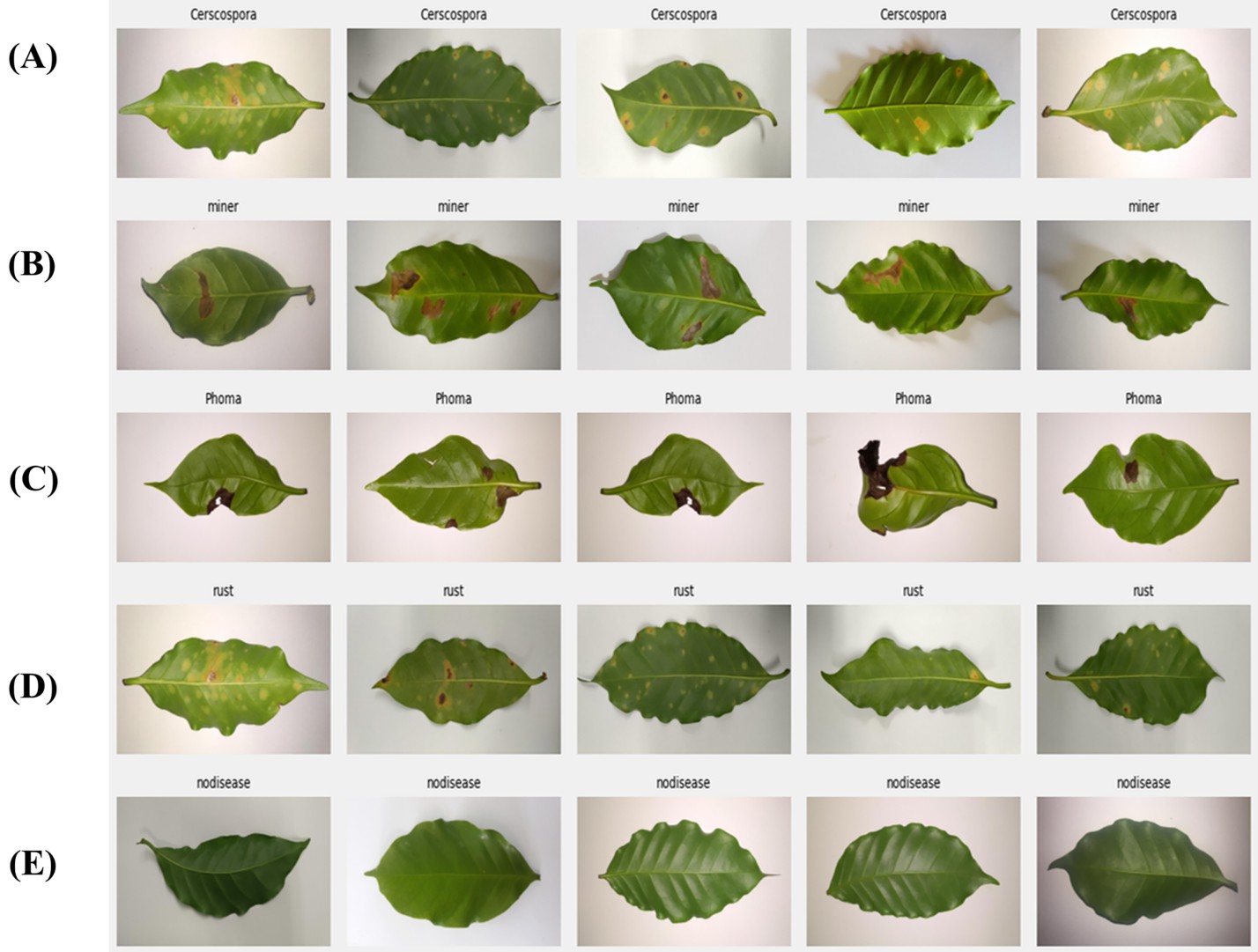

**Figure 9** Disease prediction: (A) Cerscospora, (B) Miner, (C) Phoma, (D) Rust, (E) No Disease.

of coffee disease, but most of these specialist approaches are not accessible to farmers and require the advice of experts for a correct application. CNN models have shown better performance and generalization than traditional techniques in computer vision tasks, Developing models that can be implemented on low-cost devices with the use of the cloud for precise processing capable of meeting the demand for disease detection and classification is the aim of this work, where the proposed models can be deployed in low-cost devices such as a microcontroller board or cell phone. The use of low-cost devices for deploying these models is based on several keys' factors:

- Efficiency: CNN1, CNN1 & SVM, CNN1 & RF, CNN2, CNN2 & SVM, CNN2 & RF, VGG16, VGG16 & SVM, VGG16 & RF, ResNet-50, EfficientNet) have been optimized to run efficiently on devices with limited computational power. Techniques like model

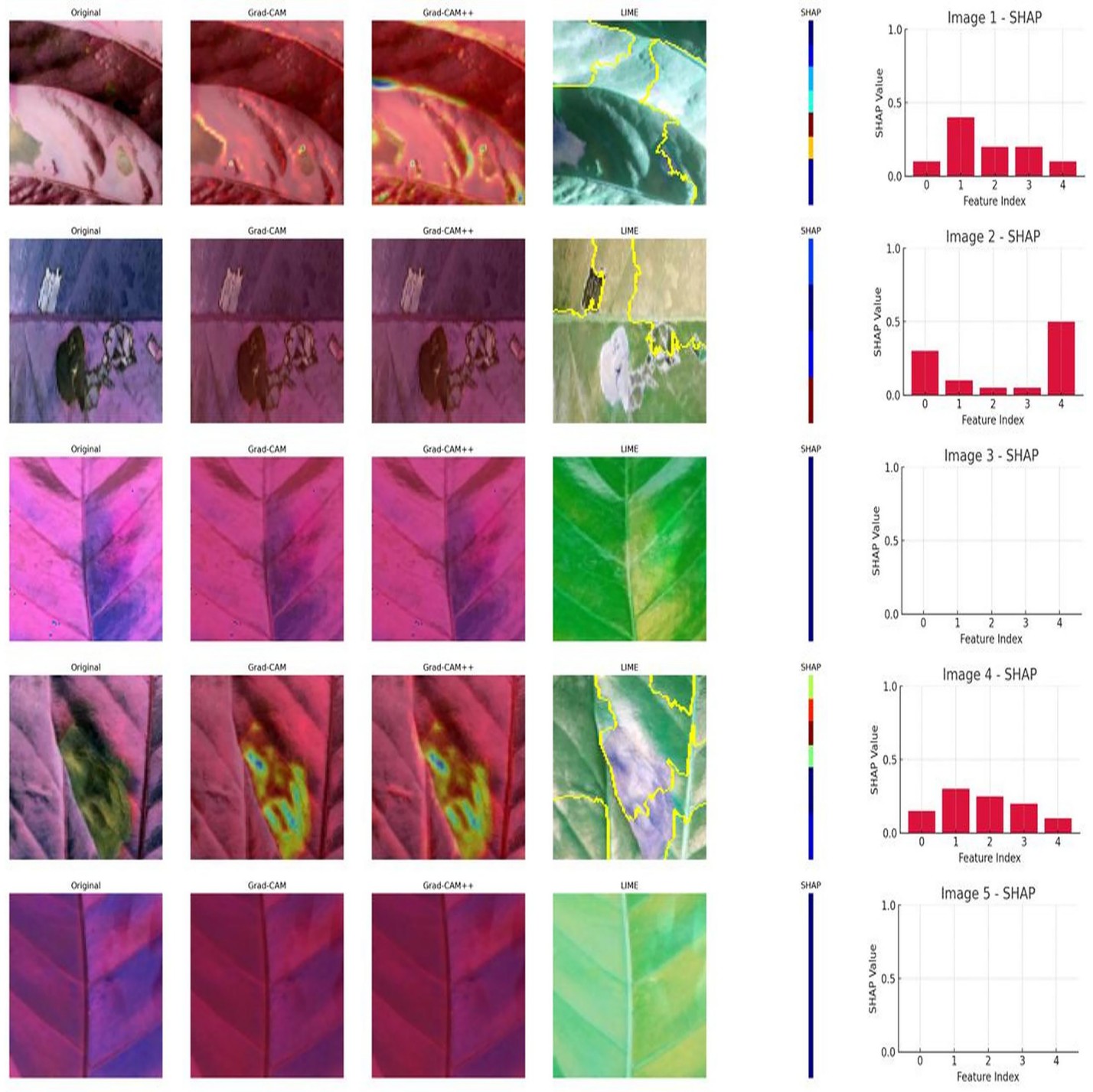

**Figure 10 Visual comparison for interpretation of Grad-CAM, Grad-CAM++, LIME, and SHAP for CNN1.**

pruning, quantization, and the use of efficient layers can significantly reduce resource requirements. The model proved effective, fast and achieved very high accuracy

- Testing time: The software's computation of the model's testing time revealed that it was remarkably low, indicating the model's efficiency. This reduced testing duration is a significant advantage, as it not only reflects the model's optimized design but also makes it highly suitable for deployment on low-cost devices, where time and resource efficiency are crucial for practical applications.
- Energy consumption: Low-cost devices typically consume less power, which is crucial for battery-operated and remote systems where power efficiency is essential.
- Flexible voting: To improve overall accuracy, a flexible voting mechanism was adopted. This method combines the outputs of all three models, allowing for more reliable and robust predictions.
- Model saving: After successful training, the models are saved for deployment.
- User interface: A user-friendly interface developed, enabling users to upload images easily and receive instant predictions. The interface is optimized for quick processing, ensuring minimal latency between image upload and prediction results.
- External camera integration: A dedicated program developed to interface with an external camera, allowing real-time image capture. Captured images are quickly processed and saved, providing a continuous stream of data that can be used for future retraining and model improvement.

## CONCLUSION AND FUTURE WORKS

In this work, we introduce a system utilizing affordable devices to aid farmers in detecting various diseases affecting coffee plants. Deep learning techniques are employed to identify and categorize these diseases, with CNNs utilized for disease classification. Through an initial hyperparameter setup, our research showcases the effectiveness of artificial neural networks in disease detection, image classification, and data processing, significantly reducing training time. Two datasets were combined, comprising over 58,000 images encompassing prevalent coffee plant diseases, including the most hazardous kinds. We proposed a new approach that introduces a multi-algorithm that classifies and selects the utmost accuracy based on deep learning models and machine learning approaches, involving convolutional neural networks (CNN1, CNN2), and transfer learning (VGG16, ResNet50, and EfficientNet). In addition to hybrid approaches, deep learning is used to deduce the features and is fed to machine learning supervised classifiers (SVM and RF). In terms of training, these 11 models were trained with a gigantic dataset to provide the reliability and ability to learn different types of patterns, which ensures providing precise results in prediction. In contrast, testing with a new dataset allows us to be sure about the outcome of the framework for DataSet-1 (99%) and DataSet-2 (98%). These achievements reflect that our proposed approach can be generalized and is particularly applicable in the real world, alongside can be relied upon as a solution and aid for farmers and researchers in detecting coffee crop diseases. This advancement empowers farmers to safeguard their resources for present and future consumers through enhanced disease diagnosis methods.

Future research should explore applying these methods to other parts of the coffee plant, like stems and roots, as well as investigating different types of coffee diseases. Additionally, examining alternative texture features, such as spectral features, is recommended for further studies. Companies are encouraged to implement this innovative approach to support farmers effectively when needed, enabling the classification of coffee diseases even on mobile devices. Future works also include exploring the potential for automating disease treatment applications on coffee plants.

## Limitations

(1) Environmental constraints or different environmental conditions such as rain or shadows may affect the quality of the images and make it difficult to distinguish between disease patterns.

(2) The model may not be able to detect diseases that were not included in the training data. Therefore, there is not enough labeled data available for all possible types of coffee diseases, which limits the model's ability to recognize all cases.

## ACKNOWLEDGEMENTS

The authors extend their sincere gratitude and appreciation to the University of Malaga, Department of Computer Engineering for providing us with access to their laboratory facilities, which were essential for conducting our research.

### Funding
The authors received no funding for this work.

### Competing Interests
The authors declare that they have no competing interests.

### Author Contributions
- Nameer Baht conceived and designed the experiments, performed the experiments, analyzed the data, performed the computation work, prepared figures and/or tables, authored or reviewed drafts of the article, and approved the final draft.
- Enrique Dominguez conceived and designed the experiments, performed the experiments, analyzed the data, performed the computation work, prepared figures and/or tables, authored or reviewed drafts of the article, and approved the final draft.
- Saif Aljumaili conceived and designed the experiments, performed the experiments, analyzed the data, performed the computation work, prepared figures and/or tables, authored or reviewed drafts of the article, and approved the final draft.

### Data Availability
The code is available in the Supplemental Files.

The data is available at Mendeley: Jepkoech, Jennifer; Mugo, David; Kenduiywo, Benson; Chebet, Edna (2021), "JMuBEN2", Mendeley Data, V1, doi: 10.17632/tgv3zb82nd.1.

## Supplemental Information

Supplemental information for this article can be found online at http://dx.doi.org/10.7717/peerj-cs.3172#supplemental-information.

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
