# Peer review of "Detection of coffee leaf disease and identification using deep learning"

_PeerJ Computer Science, doi:10.7717/peerj-cs.3172_

## Round 0.1 · original submission · Major Revisions

The manuscript Detection of Coffee Leaf Disease and Identification Using Deep Learning presents an interesting and promising approach, but several key issues need to be addressed prior to publication. A thorough major revision is required to improve overall clarity of presentation, quality of writing, technical rigor, and reproducibility.

In particular, the claim of "perfect results" is concerning, as it is not supported by appropriate statistical validation, and the obtained 99% performance might well be due to overfitting. Considering this point is of foremost importance prior to re-submission.

Please address all the detailed suggestions supplied by reviewer 1 prior to re-submission. Regarding reviewer 2’s comments, the “Additional comments” can be considered as suggestions – something to consider. Otherwise, please address the rest of the points prior to re-submission.

Finally, please ensure the language in the re-submission is clear and unambiguous, grammatically correct, and in general conforms to professional standards of scientific writing. It is recommended that you use a proofreader to ensure quality of writing prior to re-submitting.

Please include in the next submission also a version where all the figures are flowing within the text itself – this is to ensure ease of next round of review.

·

Basic reporting

1. Abstract: The term "assessment matrices" seems unclear—did you mean "evaluation metrics"?. The phrase "Our method obtained perfect results" is an overstatement. No model is perfect; instead, use a more measured statement like "achieved high performance across all evaluation metrics."

Lack of Dataset Details: The dataset consists of 58,555 images across five disease classes, but its source, balance among classes, and augmentation techniques are not mentioned.

What is the source of the dataset used for training? Was it collected manually or from an existing repository?

Vague Statement on Applicability: The final sentence, "the proposed method can be applied to detect different types of diseases affecting coffee," is too broad. Has the model been tested on unseen coffee disease types beyond the five classes mentioned?

2. Introduction: Incorrect word use in "growth of product production" (line 29) → Change to "growth of coffee production." Unnecessary repetition of "To support farmers, provide an easy-to-use and low-cost system..." at lines 76–85.

Repetitive phrasing in the Motivation section (lines 76–86): The phrase "To support farmers, provide an easy-to-use and low-cost system..." is repeated.

Were traditional machine learning classifiers (e.g., SVM, Random Forest) tested for comparison against CNN-based models?

3. Literature Review: Convert it to a table-based literature survey provides a structured and clear comparison of past research, highlighting methodologies, datasets, performance, and limitations. Below is a suggested table structure for your literature survey.

Reference, Approach/Model Used, Dataset Details, Number of Diseases Identified, Best Accuracy (%), Key Findings, Limitations

Experimental design

4. Materials and Methods:
Computational Efficiency: What was the training time and resource consumption for CNN1 vs. VGG16? Were techniques like transfer learning or quantization applied to optimize inference for mobile devices?
In feature Extraction, what texture-based features were explored, and how do they impact model interpretability?

5. Table 1: The image quality is poor, making it difficult to analyze details effectively. Enhancing resolution and clarity is necessary for better interpretation.

Table 5: The results appear unrealistic as they have not been compared with real-time images. While the reported accuracy is high, it raises concerns about the reliability of the proposed algorithm. A more rigorous evaluation with real-world data is needed to validate the claimed performance.

Figure 1: The experimental flowchart lacks visual clarity and appears dull. Enhancing its quality with better resolution, contrast, and design elements would improve readability and presentation.

Figures 2, 4, and 5: The graphs appear too basic and do not meet standard presentation quality. Refining them with improved formatting, clearer labels, and a more professional design would enhance their readability and impact.

Validity of the findings

6. Conclusion: How feasible is it to extend this approach to mobile devices while maintaining high accuracy and low computational costs?

7. Reference: The Author cited old papers (2019, 2020, 2021), and also, they are irrelevant; change them to recent papers.

·

Basic reporting

1. The paper should ensure clear and precise terminology. Can the authors refine the abstract to improve readability and eliminate redundancies (e.g., "proliferation of coffee-related diseases Signiant impacts" should be revised for clarity)?

2. The manuscript should include proper citations for all state-of-the-art models used for comparison.

3. Clearly state the specific deep learning models used in the abstract and introduction.

4. Provide a structured description of the dataset, including image sources, annotation process, and preprocessing techniques.

5. The conclusion mentions "three key algorithms for comparison" but does not explicitly list them. Can the authors explicitly specify these algorithms for better clarity?

Experimental design

1. Specify the architecture details of the CNN models, including layer configurations, activation functions, and optimizer settings.

2. Justify the choice of dataset size and distribution—how well does the dataset represent real-world conditions?

3. The dataset consists of 58,555 images across five classes. Can the authors provide a detailed breakdown of dataset composition, including class distributions and any augmentation techniques used?

4. Explain the rationale behind the hyperparameter settings and training configurations (e.g., learning rate, batch size, number of epochs).

5. Provide insights into the computational resources used for training, such as GPU specifications.

6. Clarify if data augmentation techniques were applied and how they influenced model performance.

7. Explain why CNN1 and VGG16 were chosen for comparison and whether other architectures (e.g., ResNet, EfficientNet) were considered.

8. Include a comparative evaluation against multi-crop disease detection models to highlight the strengths and weaknesses of the proposed approach.

9. Were all images collected from the same geographical region, or does the dataset incorporate data from multiple locations? If not, how do the authors ensure model generalizability?

10. The paper mentions that "cost-effective devices" were used. Can the authors provide specific details on the hardware specifications and computational efficiency of the proposed system?

Validity of the findings

1. Report confidence intervals or statistical significance tests to validate performance improvements. Have the authors performed cross-validation to ensure robustness, and if so, which method (e.g., k-fold cross-validation) was used?

2. Discuss generalizability—can the model detect coffee leaf diseases across different regions and environmental conditions?

3. The study claims 99% accuracy. Given the high performance, did the authors analyze the potential impact of dataset biases? Is there any risk of overfitting due to dataset imbalance?

Additional comments

1. Suggest exploring alternative feature extraction techniques, such as spectral features or hyperspectral imaging.

2. The authors should consider extending the study to include comparisons with multi-crop disease detection models.

3. The manuscript should provide insights into potential real-world applications. Can the model be optimized for deployment on edge devices such as smartphones for real-time disease detection?

4. Discuss real-world deployment feasibility, including edge computing or mobile-based inference.

5. Recommend integrating explainability techniques (e.g., Grad-CAM, SHAP) to interpret model decisions.

6. Suggest expanding the dataset to include stem and root diseases for a more comprehensive analysis as future work.

---

## Round 0.2 · accepted · Accept

The reviewer seems satisfied with the recent changes and therefore I can recommend this article for acceptance.

·

Basic reporting

-

Experimental design

-

Validity of the findings

-